# Minimax Bayesian Neural Networks

**DOI:** 10.3390/e27040340

**Published:** 2025-03-25

**Authors:** Junping Hong, Ercan Engin Kuruoglu

**Affiliations:** Tsinghua Shenzhen International Graduate School, Tsinghua University, Shenzhen 518055, China; hjp23@mails.tsinghua.edu.cn

**Keywords:** Bayesian neural networks, robustness, noise perturbation, minimax game, closed-loop neural networks, maximal coding rate distortion

## Abstract

Robustness is an important issue in deep learning, and Bayesian neural networks (BNNs) provide means of robustness analysis, while the minimax method is a conservative choice in the classical Bayesian field. Recently, researchers have applied the closed-loop idea to neural networks via the minimax method and proposed the closed-loop neural networks. In this paper, we study more conservative BNNs with the minimax method, which formulates a two-player game between a deterministic neural network and a sampling stochastic neural network. From this perspective, we reveal the connection between the closed-loop neural and the BNNs. We test the models on some simple data sets and study their robustness under noise perturbation, etc.

## 1. Introduction

Nowadays, deep learning, as a data-driven method, has become more and more popular and has been applied to multiple areas, such as weather forecasting [1] and image classification [2,3]. Most neural networks are trained with supervised learning with an end-to-end framework. Representation learning seeks a good representation of the trained data, such as learning representation by mutual information [4] and maximal coding rate [5].

Although deep learning seems to be rather successful, robustness issues still haunt the field of deep learning [6,7]. Mackay [8] discussed the issues about determinist neural networks and introduced the first framework of Bayesian neural networks (BNNs); Neal [9] proposed Monte Carlo Monte Carlo (MCMC) methods for the implementation of BNNs which are computationally intensive. Dropout is a well-known technique that is proposed to prevent overfitting [10] and can be seen as an approximation of Bayesian methods [11]. BNNs aim to learn a distribution of neural networks through posterior estimation and use random variables to describe the weights of neural networks and update the mean and the variance simultaneously [12]. Previous works have shown that BNNs can quantify the uncertainty of neural networks [13], are robust to the choice of prior [14], and are more robust to gradient attack than deterministic neural networks [15]. For Prior Networks, Malinin and Gales [16] argue that the randomness of deep learning includes model uncertainty, data uncertainty, and distributional uncertainty, and people can use Prior Networks to perform out-of-distribution detection. Wang et al. [17] have applied MCMC methods to the information bottleneck study. In addition, the minimax method is often thought of as a robustness help for Bayesian methods [18], and it will improve the robustness at the cost of accuracy because it considers the best case of the worst case.

The minimax method has been used for a long time. Previous studies using the minimax game to study the robustness of neural networks use fault-tolerant neural networks [19,20,21] and obtain the two-player game between normal neural networks and fault neural networks. Closed-loop transcription neural networks [22] design a new two-player game between the decoder and the composition of encoder and decoder with a minimax loss for representation learning.

Inspired by the previous minimax work [22] and the classical BNNs, we apply the minimax game to the classical BNNs. The reason for this is to make the BNNs more conservative and reveal the connection between the closed-loop neural networks and BNNs. To the best of our knowledge, this is the first time that the classical BNNs have been separated as two neural networks through the minimax game. Other minimax works, such as fault-tolerant neural networks [21], care about how the prediction performance of neural networks behaves under different levels of fault nodes or edges in neural networks, which is similar to dropout techniques. Compared with closed-loop transcription networks [22], they use the composition of some deterministic neural networks to obtain the minimax game, while we use a stochastic neural network instead.

The contribution of this paper includes two perspectives. One is that we propose a naturally more conservative variant of a BNN and we can adjust the variance level of the weights of BNNs through minimax loss. And our framework can provide a reference for the variance setting for the classical BNNs. The other is that our framework reveals the connection between closed-loop neural networks and BNNs. In addition, this formulation provides flexibility to change the suitable randomness or variance for the noise part.

The paper is organized as follows: Section 2 introduces the formulation of the minimax BNNs. Section 3 presents the experiments and results. Section 4 is a discussion of the research.

## 2. Minimax Bayesian Neural Networks

Maximal coding rate reduction (MCR) first appeared as a loss for representation learning in 2020 [5]. Then, closed-loop neural networks formulated a minimax loss with MCR [22]. Minimax Bayesian neural networks (BNNs) use the same minimax loss as [22], and the main difference is that our minimax game is between a deterministic neural network *f* and a random sampling neural network g=f+r∗ξ, where *r* is the radius of the hypersphere controlling the variance, and ξ is random noise. Besides, if we view the classical BNNs as a hypersphere, then *f* is the center and *g* is sampled on the equator. The minimax formulation for BNNs is given by(1)$$\begin{array}{c} \min_{g}\max_{f}\tau \left(f,g\right)≐\Delta R\left(f\left(X\right)\right)+\Delta R\left(g\left(X\right)\right)+\Delta R\left(f\left(X\right),g\left(X\right)\right)\\ =\Delta R\left(Z\right)+\Delta R\left(\hat{Z}\right)+\Delta R\left(Z,\hat{Z}\right), \end{array}$$
where *X* denotes the data, *f* denotes a deterministic neural network, and *g* denotes the sampling stochastic neural network. ΔR(f(X)) and ΔR(g(X)) denote the data compression with different labels by *f* and *g* through the MCR loss, and ΔR(f(X),g(X)) characterizes the compression difference by *f* and *g* for the same data set. For the rigorous definition, please see [22]. Z=f(X) denotes the final output of representation learning for *f*, and $\hat{Z}=g\left(X\right)$ denotes the same for *g*. Note that the third term of this loss controls the gap between *f* and *g*. The current setting allows the gap to be relatively large sometimes, so we also use the log case to denote a smaller gap as $log(1+\Delta R\left(Z,\hat{Z}\right))$, one example is shown in Section A.2.

Because MCR loss might not be very straightforward, here we provide an equivalent formulation via supervised learning. Noting that the main focus is to understand the formulation of the minimax BNNs and reveal their connection to the closed-loop neural networks, we only run our experiments via representation learning formulation.(2)$$\begin{array}{c} \min_{f,g}\tau \left(f,g\right)≐loss\left(f\left(X\right)\right)+loss\left(g\left(X\right)\right)\\ \;s.t.\;\begin{array}{c} pre(f(X))-pre(g(X))=c, \end{array} \end{array}.$$
where *X*, *f*, and *g* denote the same as the previous case. loss(f(X)) and loss(g(X)) represent the losses for *f* and *g*, such as cross-entropy for classification. pre(f(X)) and pre(g(X)) are the final predictions for *f* or *g*. *c* is a constant that denotes the gap between *f* and *g*, for example, *c* could be 100 for 1000 predictions, which means we allow the maximal difference between *f* and *g* to be 10 percent. Note that we can use the Lagrange method to transform this into a min-max or max-min formulation [21].

Compared with the classical BNNs, which represent only one realization, the minimax Bayesian framework will train two neural networks, one is the mean or the center point and the other is sampled on the equator. Because minimax BNNs consider both the best case and the worst case, they are more robust than the classical BNNs, with the cost of performance due to the perturbation in *g*. Another difference is that the variance of weights is given in advance and updated across the training process, while the variance level of minimax BNNs is given by the setting gap and updated by the sampling. Then, for closed-loop neural networks with the same minimax loss as the representation in [22], the two deterministic neural networks are *f* and f∘g∘f. The main issue is that f∘g∘f cannot change as quickly as g=f+r∗ξ, since we only need to find a suitable level for *r*. In their formulation, they need to add multiple activation functions to support the image generation, hence they need to use the batch normalization (BN) layer to accelerate the process [23].

## 3. Experiments and Results

The data sets include MNIST [24], Fashion MNIST (FMNIST) [25], and CIFAR-10 [26]. For MNIST and FMNIST, we use the same convolutional neural network (CNN) [27] as [22]. The optimization algorithm is Adam(0.5, 0.999), and the learning rate is 0.001. *f* only has one kind of activation function with Leakyrelu; *f* is initialized with N(0,0.02) and ξ is sampled from N(0,0.01), and this might change if we are required to update the shape of ξ with Bayes by Backpropagation [13]. Normally, the zone for gold search is from 0 to 100, and we will use grid search to validate our correctness. After training, we map the data to the subspace and use the k-nearest neighbors (KNN) method [28] to predict the labels implemented through the scikit-learn package [29]. In addition, the suitable radius *r* is from 0.2 to 0.5 for the log case on MNIST data, and the *r* will decrease across the training process; *r* is usually about 3 to 6 without changing to the third term. We build the model through PyTorch (https://pytorch.org/, accessed on 13 March 2025) [30], and the codes are public at https://github.com/Jacob-Hong17/MinMax-BNN, accessed on 30 December 2024.

### 3.1. Main Results

In Table 1, we can see that the results of minimax BNNs are slightly worse than the closed-loop work in most cases because our formulation is also minimax, and the component neural network will bring more noise into the training process. However, our formulation is more robust than the closed-loop work because in every sampling we can always obtain the suitable *r* by sampling, while f∘g∘f can not quickly update like g=f+r∗ξ. The closed-loop idea is of great importance in the control theory; here, we reveal the connection between the closed-loop work and the minimax BNNs. Apart from these findings, we also study the meaning of *r* for the MCR loss and some issues for searching for the suitable *r* in Appendix A, see Figure A1, Figure A2, Figure A3 and Figure A4.

In Table 2, we can see the best sampling result is for r=0.2, this is because our formulation is minimax, which considers both the best case and the worst case. This is the reason why minimax BNNs can only obtain good results rather than the best results. However, this formulation is easy to implement, and the sampling results after training can be used as a reference for classical BNNs. For these results, classical BNNs should use a hyperparameter of variance that is at least below 0.2.

### 3.2. Noise Perturbation

Furthermore, we test how *r* changes when adding noise to the data. In this part, we set the corruption ratio as 0,0.05,0.1,0.2,0.3,0.4,0.5,0.6,0.7,0.8,0.9 with Gaussian random noise and normalization of the data and find the corresponding radius, see Figure 1. As we can see, the radius will normally decrease and then increase if we enlarge the noise ratio from 0 to 1. The reason why *r* will drop in the beginning is because adding small noise is equivalent to small perturbations from the Taylor expansion perspective. We can see that *r* will usually increase after reaching a changing point. Because there is more and more noise in the data, this requires more perturbation on *g* to obtain the fixed gap requirements based on the MCR loss.

### 3.3. Dataset Similarity

Our formulation for out-of-distribution detection is very similar to the one in [31] because A(X+ξ)+b=(A+ξ)X+b. Adding noise to the neural network is usually more costly, but our formulation allows us the flexibility to adjust the noise levels from small to large in contrast to the classical BNNs or previous work. In Table 3 and Table 4, we generate the corresponding radius *r* for different data sets to evaluate their similarity and obtain results similar to those in [31,32]. These clearly show that similar data sets will have a lower *r* or compression volume based on the trained neural networks *f*, and non-Gaussian noise like Cauchy noise will have a larger volume than in the Gaussian case.

## 4. Discussion

In this paper, we apply the minimax game to classical BNNs and obtain a more conservative variant of a BNN. Furthermore, we reveal the connection between closed-loop neural networks and our minimax BNNs and point out the limitation that closed-loop neural networks are not able to quickly adapt to the loss requirement compared with the minimax BNNs. Last but not the least, this framework provides some reference for variance setting for the classical BNNs and is more flexible in adjusting the variance level for well-trained models.

One limitation of this paper is that the distribution of the random sampling neural networks is always a Gaussian distribution, and we believe that minimax BNNs with non-Gaussian distributions might be worth exploring.

## Figures and Tables

**Figure 1 entropy-27-00340-f001:**
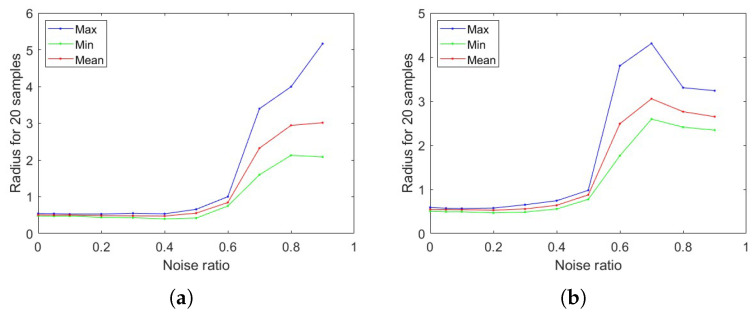
Perturbation impact on different data sets: (**a**) MNIST, (**b**) FMNIST.

**Table 1 entropy-27-00340-t001:** Comparison of accuracies of Minimax BNN and Closed-Loop NNs.

Models	*f* (MNIST)	Closed-Loop	*f* (FMNIST)	Closed-Loop
test 1	96.28%	96.57%	85.82%	86.09%
test 2	96.43%	96.40%	85.79%	86.24%
test 3 (log)	96.70%	96.82%	86.21%	86.81%
test 4 (log)	96.73%	97.25%	86.09%	86.71%

**Table 2 entropy-27-00340-t002:** Accuracy of sampling neural networks *g* on MNIST.

*r* (Test 4, 20 Samples)	Max	Min	Mean	Variance
0.1	97.01%	96.93%	96.97%	5.0 × $10^{-8}$
0.2	97.07%	96.9%	96.96%	1.5 × $10^{-7}$
0.5	97.03%	96.84%	96.93%	2.6 × $10^{-7}$
1	96.95%	96.65%	96.81%	7.1 × $10^{-7}$
2	96.38%	95.18%	95.82%	9.8 × $10^{-6}$
3	92.21%	78.3%	84.78%	1.7 × $10^{-3}$
4	62.99%	26.33%	40.83%	9.7 × $10^{-3}$
6	17.88%	10.45%	13.33%	5.0 × $10^{-4}$
8	14.54%	8.22%	10.84%	1.8 × $10^{-4}$
10	11.93%	7.91%	10.29%	1.2 × $10^{-4}$

**Table 3 entropy-27-00340-t003:** Different *r* values for other data sets trained by MNIST.

r (log, 20 Samples)	Max	Min	Mean	Variance
MNIST	0.544	0.481	0.507	3.1 × $10^{-4}$
FMNIST	1.768	1.186	1.473	1.8 × $10^{-2}$
CIFAR-10 (channel 1)	3.134	1.999	2.639	1.0 × $10^{-1}$
Gaussian	3.751	1.778	2.661	3.1 × $10^{-1}$
Laplace	5.637	2.902	3.800	4.9 × $10^{-1}$
Cauchy	5.807	4.188	4.942	1.9 × $10^{-1}$

**Table 4 entropy-27-00340-t004:** Different *r* values for other data sets trained by FMNIST.

r (log, 20 Samples)	Max	Min	Mean	Variance
FMNIST	0.591	0.503	0.546	5.4 × $10^{-4}$
MNIST	1.904	1.479	1.671	1.6 × $10^{-2}$
CIFAR-10 (channel 1)	2.599	2.155	2.458	1.2 × $10^{-2}$
Gaussian	3.220	2.189	2.560	7.6 × $10^{-2}$
Laplace	4.129	2.852	3.487	9.4 × $10^{-2}$
Cauchy	5.851	4.121	4.723	1.6 × $10^{-1}$

## Data Availability

Data is contained within the article.

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
