# Peer review of "Minimax Bayesian Neural Networks"

_entropy, 2025, doi:10.3390/e27040340_

Round 1

Reviewer 1 Report

Comments and Suggestions for Authors

The paper addresses robustness of deep learning.
The authors consider Bayesian neural networks and
apply a game theoretic concept, the minimax approach,
to obtain robust results. Specifically, they consider a
game between deterministic neural networks and stochastic
neural networks and then use the minimax criterion
(or a related constraint minimization criterion). The price
they have to pay for the robustness gain is that they
have to train two neural networks. They illustrate the
usefulness of their approach for various data sets and
noise perturbations of the data.

I think the paper falls into the scope of the journal
entropy. In fact, in the description of the scope of the
journal, the topics neural networks and game theory are
explicitly mentioned. The numerical examples in the paper
suggest that the present approach can be an attractive
alternative to classical Bayesian neural networks.

Some minor suggestions:

I believe the first paragraph of Section 2 is not so easy
to understand without consulting [23]. Perhaps a few words
could be added so that a reader can directly understand these
ideas. Also, a small numerical example could be added to
demonstrate the "pretty large gap", mentioned at the end of
the paragraph.

The idea of Table 1 is to demonstrate that the present
minimax results are only slightly worse than the closed-loop
results (if one disregards the advantage of robustness).
The comparison is fine for the MNIST data set, but for
the FMNIST data set, there are no entries in table for the
closed-loop approach. Would it be possible to include them
for an additional comparison?

Author Response

Comment 1: The paper addresses robustness of deep learning.
The authors consider Bayesian neural networks and
apply a game theoretic concept, the minimax approach,
to obtain robust results. Specifically, they consider a
game between deterministic neural networks and stochastic
neural networks and then use the minimax criterion
(or a related constraint minimization criterion). The price
they have to pay for the robustness gain is that they
have to train two neural networks. They illustrate the
usefulness of their approach for various data sets and
noise perturbations of the data.

I think the paper falls into the scope of the journal
entropy. In fact, in the description of the scope of the
journal, the topics neural networks and game theory are
explicitly mentioned. The numerical examples in the paper
suggest that the present approach can be an attractive
alternative to classical Bayesian neural networks.

Some minor suggestions:

I believe the first paragraph of Section 2 is not so easy
to understand without consulting [23]. Perhaps a few words
could be added so that a reader can directly understand these
ideas. Also, a small numerical example could be added to
demonstrate the "pretty large gap", mentioned at the end of
the paragraph.

Respond 1: Thanks for your valuable comments, we agree with your kind suggestion.  For the first suggestion, we add some words to explain the loss from Lines 68 to 71, Section 2. For the second suggestion, the words are changed to " The current setting allows the gap to be relatively large sometimes, so we also use the log case to denote a smaller gap ..., one example is shown in Appendix A.2.", see that from Line 73 to 75.  

Comment 2: The idea of Table 1 is to demonstrate that the present
minimax results are only slightly worse than the closed-loop
results (if one disregards the advantage of robustness).
The comparison is fine for the MNIST data set, but for
the FMNIST data set, there are no entries in table for the
closed-loop approach. Would it be possible to include them
for an additional comparison?

Respond 2: Thanks for your kind advice, we agree that and implement the closed-loop method in FMNIST data, and add the results in Table 1.

Besides, we also improved the language throughout the paper, please see the attachment.

Reviewer 2 Report

Comments and Suggestions for Authors

General remarks:
The subject is of interest for Bayesian NN readers. The paper is well written. However, following some rules of edition is needed. Section 2 can be slightly extended for introducing the main subject. The position of the paper with respect to the references 5, 21, 23 have to be more explained to appreciate the novelty of the paper.
The captions of the tables and figure can be more explicit.
Section Discussions has to be more extended.

Some detailed coments are on the pdf file of the paper.

Comments on the Quality of English Language

Small editing is required

Author Response

General remarks:
The subject is of interest for Bayesian NN readers. The paper is well written. However, following some rules of edition is needed. Section 2 can be slightly extended for introducing the main subject. The position of the paper with respect to the references 5, 21, 23 have to be more explained to appreciate the novelty of the paper.
The captions of the tables and figure can be more explicit.
Section Discussions has to be more extended.

Some detailed comments are on the pdf file of the paper.

Comments 1: However, following some rules of edition is needed. 

Respond 1:  Thanks for your comprehensive and professional comments, we update the paper by your attachment, see that in the attachment.

Comments 2: Section 2 can be slightly extended for introducing the main subject.

Respond 2: We revised Section 2 from Line 60 to 75.

Comments 3:  The position of the paper with respect to the references 5, 21, 23 have to be more explained to appreciate the novelty of the paper.

Respond 3: In the previous version, paper 21 and paper 23 are both minimax work in neural networks as minimax BNNs, and their main difference is the formulation of the second player, we added more explanation to tell their difference in the updated version from line 41 to line 50. Paper 5 proposes the MCR loss in representation learning and later paper 23 uses that loss to build the minimax game. We explained that in the new version of Section 2, lines 60 to 66. Note that paper 23 becomes paper 22 in the new version because we reduced 2 references.

Comments 4: The captions of the tables and figure can be more explicit.

Respond 4: We update the captions of Table 2, Figure 1, and others areas that your attachment has pointed out.

Comments 5: Section Discussions has to be more extended.

Respond 5: We revised the discussion, and added limitations and future work from line 141 to line 150,  Section 4.

Comments 6: Some detailed coments are on the pdf file of the paper.

Respond 6: We revised the position with a yellow label, like the indent issue and MCR definition issues in Section 2, and KNN with definition in Section 3, and change "Pythorch" to "PyTorch".   

Comments 7: Small editing is required.

Respond 7: We revised the language throughout the paper, see the attachment. Again, thanks for your exhaustive and useful advice.

Round 2

Reviewer 1 Report

Comments and Suggestions for Authors

I am satisfied with the revision.

Author Response

Comments 1: I am satisfied with the revision.

Response 1: Thanks for your appreciation and previous advice, we improved the language throughout the paper. 

Reviewer 2 Report

Comments and Suggestions for Authors

The paper is now almost good. Yet, the authors have to check the equation 1, as the index i is missing. See some other comments on the pdf file concerning the references, Figures and Tables.

Author Response

Comments 1: The paper is now almost good. Yet, the authors have to check the equation 1, as the index i is missing. See some other comments on the pdf file concerning the references, Figures and Tables.

Response 1: Many thanks for your comprehensive and professional advice. For the equation 1, we deleted the index i in equation 1 and related areas in Line 74. For the references, we revised the related area in yellow, see that in the attachment. For Figures and Tables, we also updated the related words labeled in yellow, added one conclusion in Lines 131-136, Section 3.2, and added another conclusion in Lines 143-145, Section 3.3. In addition, we revised the language throughout the paper to make it more fluent, especially the places labeled in yellow.  See that in the attachment. Again, thanks for your valuable suggestion and hard work. 

Round 3

Reviewer 2 Report

Comments and Suggestions for Authors

The manuscript is now acceptable